Lateralization (handedness) in Magellanic penguins

Stor Thaís 1
Rebstock Ginger A. 2 3 gar@uw.edu
García Borboroglu Pablo 2 3 4
Boersma P. Dee 2 3 5
1 Centro de Ciências Biológicas, Departamento de Ornitologia, Universidade Federal de Pernambuco , Recife, Pernambuco , Brazil
2 Center for Ecosystem Sentinels, Department of Biology, University of Washington , Seattle, WA , USA
3 Global Penguin Society , Puerto Madryn, Chubut , Argentina
4 CESIMAR, CCT—CENPAT—CONICET , Puerto Madryn, Chubut , Argentina
5 Wildlife Conservation Society , Bronx, NY , USA
Rogers Lesley
Electronic publication date: 2019 May 20
Publication date: 2019
Volume: 7
Electronic Location ID: e6936
Received 2018 Oct 11; Accepted 2019 Apr 10
Copyright: © 2019 Stor et al.
Copyright year: 2019
Copyright holder: Stor et al.
License: This is an open access article distributed under the terms of the Creative Commons Attribution License, which permits unrestricted use, distribution, reproduction and adaptation in any medium and for any purpose provided that it is properly attributed. For attribution, the original author(s), title, publication source (PeerJ) and either DOI or URL of the article must be cited.
License URL: https://creativecommons.org/licenses/by/4.0/

Keywords: Aggression, Behavior, Fight orientation, Footedness, Handedness, Lateralization, Magellanic penguin, Sex, Spheniscus magellanicus, Thermoregulation

Funding: Wildlife Conservation Society (WCS), Exxonmobil Foundation, the Pew Fellows Program in Marine Conservation, the Disney Worldwide Conservation Fund, the National Geographic Society, the Chase, Cunningham, MKCG, Offield, Peach, Thorne, Tortuga, and Kellogg Foundations, the Wadsworth Endowed Chair in Conservation Science, and Friends of the Penguins Research at Punta Tombo was funded by the Wildlife Conservation Society (WCS), Exxonmobil Foundation, the Pew Fellows Program in Marine Conservation, the Disney Worldwide Conservation Fund, the National Geographic Society, the Chase, Cunningham, MKCG, Offield, Peach, Thorne, Tortuga, and Kellogg Foundations, the Wadsworth Endowed Chair in Conservation Science, and Friends of the Penguins. There was no additional external funding received for this study. The funders had no role in study design, data collection and analysis, decision to publish, or preparation of the manuscript.

==============================
Lateralization, or asymmetry in form and/or function, is found in many animal species. Brain lateralization is considered adaptive for an individual, and often results in “handedness,” “footedness,” or a side preference, manifest in behavior and morphology. We tested for lateralization in several behaviors in a wild population of Magellanic penguins Spheniscus magellanicus breeding at Punta Tombo, Argentina. We found no preferred foot in the population (each penguin observed once) in stepping up onto an obstacle: 53% stepped up with the right foot, 47% with the left foot (n = 300, binomial test p = 0.27). We found mixed evidence for a dominant foot when a penguin extended a foot for thermoregulation, possibly depending on the ambient temperature (each penguin observed once). Penguins extended the right foot twice as often as the left foot (n = 121, p < 0.0005) in 2 years when we concentrated our effort during the heat of the day. In a third year when we observed penguins early and late in the day, there was no preference (n = 232, p = 0.59). Penguins use their flippers for swimming, including searching for and chasing prey. We found morphological evidence of a dominant flipper in individual adults: 60.5% of sternum keels curved one direction or the other (n = 76 sterna from carcasses), and 11% of penguins had more feather wear on one flipper than the other (n = 1217). Right-flippered and left-flippered penguins were equally likely in both samples (keels: p = 0.88, feather wear: p = 0.26), indicating individual but not population lateralization. In fights, aggressive penguins used their left eyes preferentially, consistent with the right side of the brain controlling aggression. Penguins that recently fought (each penguin observed once) were twice as likely to have blood only on the right side of the face (69%) as only on the left side (31%, n = 175, p < 0.001). The proportion of penguins with blood only on the right side increased with the amount of blood. In most fights, the more aggressive penguin used its left eye and attacked the other penguin’s right side. Lateralization depended on the behavior tested and, in thermoregulation, likely on the temperature. We found no lateralization or mixed results in the population of Magellanic penguins in three individual behaviors, stepping up, swimming, and thermoregulation. We found lateralization in the population in the social behavior fighting.

Introduction

Asymmetry in form and/or function, known as lateralization, is common in bilaterally symmetrical animals, including fish, amphibians, reptiles, birds, mammals, and a growing list of invertebrates (Bisazza, Rogers & Vallortigara, 1998; Frasnelli, Vallortigara & Rogers, 2012; Rogers, 2017; Vallortigara, Chiandetti & Sovrano, 2011). Brain lateralization is hypothesized to be adaptive, especially in social or group behaviors (Bisazza et al., 2000; Bisazza, Rogers & Vallortigara, 1998; Ghirlanda, Frasnelli & Vallortigara, 2009; Ghirlanda & Vallortigara, 2004; Rogers, 2017). Brain lateralization appears to enhance cognitive ability in an individual by allowing both brain hemispheres to specialize and function simultaneously (Rogers, 2017), with less neural tissue (Ghirlanda & Vallortigara, 2004). It may also shorten reaction times (Dadda, Koolhaas & Domenici, 2010) and prevent conflicting responses from the two hemispheres when the eyes are not seeing the same scene (Ghirlanda & Vallortigara, 2004). Brain lateralization is often linked to behavioral lateralization, or to hand, foot, or paw preferences (Rogers, 2009). The strength of lateralization may be more adaptive than its direction within an individual. In some species, lateralized individuals forage more efficiently, escape more rapidly, or make fewer motor errors regardless of the direction of lateralization (Bell & Niven, 2016; Dadda, Koolhaas & Domenici, 2010; Ghirlanda & Vallortigara, 2004; Kurvers et al., 2017). In addition, strength of lateralization can be inherited independently of direction (Bisazza, Rogers & Vallortigara, 1998). We tested for lateralization in a wild population of Magellanic penguins Spheniscus magellanicus (Forster, 1781) at a breeding colony in Argentina.

Brain lateralization is remarkably consistent among a wide range of taxa (Ghirlanda & Vallortigara, 2004; Rogers, 2017; Vallortigara & Rogers, 2005). For example, the left hemisphere (which receives information from the right eye) controls approach and feeding behaviors including prey discrimination, capture, and manipulation, whereas the right hemisphere controls avoidance behaviors, fear, and predator detection and escape behaviors (Vallortigara & Rogers, 2005). Aggression and intense emotions are inhibited by the left hemisphere and promoted by the right (Bisazza, Rogers & Vallortigara, 1998). The left hemisphere functions in recognition of broad categories and local landmarks and cues, and moderates or even delays responses (Rogers, 2008). The right hemisphere functions in individual recognition and spatial cognition, and rapid responses that are typical of the species (Rogers, 2008).

Lateralization can occur within the population or within individuals. When more than 50% of the individuals in a population are lateralized in the same direction it is referred to as “population-level lateralization” (Bisazza, Rogers & Vallortigara, 1998), “handedness” (Vallortigara, 2015), or “directional asymmetry” (Vallortigara & Rogers, 2005). When at least some individuals in the population are lateralized, but with equal numbers in each direction, it is referred to as “individual-level lateralization” (Bisazza, Rogers & Vallortigara, 1998), “hand preference” (Vallortigara, 2015), or “antisymmetry” (Vallortigara & Rogers, 2005). Individual lateralization can occur without lateralization of the population. The brains of most animals studied show some lateralization in individuals (Rogers, 2017), but not all populations show a consistent hand, foot, paw, or side preference (Rogers, 2009; Vallortigara, 2015). Testing for lateralization within individuals requires multiple observations for each individual; testing for lateralization in the population requires one observation for each individual in a sample of the population.

Side preferences, manifested by behavior and morphology, are common even in animals without hands or paws, such as marine mammals (MacNeilage, 2014) and fish (Bisazza et al., 2000). Marine mammal populations often show strong lateralization in natural feeding behaviors, such as lunge feeding and beaching while herding fish, usually biased towards the right side. In addition, right flipper bones are larger than left flipper bones in some dolphins and porpoises (MacNeilage, 2014). Individuals in many species of fish, especially schooling fish, preferentially turn the same direction, which may function to coordinate movements in a school (Bisazza et al., 2000; Domenici, 2010).

Side preferences in individuals (individual lateralization) depend on a variety of factors, including the task or the behavioral context, age, physiological condition, and environmental factors (Versace & Vallortigara, 2015). In humans, Huurnink et al. (2014) concluded that “there is no such thing” as a dominant leg because the preferred leg depended on the task. Human volunteers used the same leg more often in skilled than in unskilled tasks and foot preference in humans can change during childhood (Schneiders et al., 2010). Lateralization likewise changes with age in some birds and rodents (Bisazza, Rogers & Vallortigara, 1998). In dogs, the preferred paw depended on the task, but was stable over 6 months within individuals (Wells et al., 2018). The preferred foot used by budgerigars Melopsittacus undulatus also varied with the task (Schiffner & Srinivasan, 2013). Herring Clupea harengus in a school had more uniform escape directions than solitary individuals (Domenici, 2010). In other fish species, environmental factors such as oxygen levels, water temperature, and acclimation temperature affected the directionality of escape responses (Domenici, 2010).

Population lateralization also depends on the context, especially whether the behavior is individual or social (Ghirlanda, Frasnelli & Vallortigara, 2009; Ghirlanda & Vallortigara, 2004). Schooling fish and social insects are more likely to show population lateralization than solitary fish and insects (Bisazza et al., 2000; Frasnelli, Vallortigara & Rogers, 2012). Even within a species, population lateralization is more likely in social behaviors than in individual behaviors (Domenici, 2010; Rogers, Frasnelli & Versace, 2016).

More observations of natural behaviors in wild populations are needed to confirm the prevalence of lateralization found in laboratory studies. In particular, more studies of the same species in a variety of behavioral contexts will provide a more complete understanding of lateralization (Versace & Vallortigara, 2015). Penguins are easily observed in breeding colonies, alone and interacting with other penguins. We looked for lateralization of individuals and whether the lateralization was apparent for the population of Magellanic penguins at Punta Tombo, Argentina. We examined behaviors in a variety of contexts: stepping up while walking (individual behavior), extending a foot for thermoregulation (individual behavior), swimming (individual behavior), and fighting (social behavior).

Dominant foot

We directly observed behavior on land to detect a dominant foot when a penguin stepped up and when a penguin was prone in a thermoregulatory posture with a foot extended. We used two tests because the dominant limb often depends on the task (Huurnink et al., 2014; Wells et al., 2018).

Individuals may use a dominant foot preferentially in stepping up onto an obstacle (Huurnink et al., 2014; Tomkins, Thomson & McGreevy, 2010). Penguins do not balance on one leg as some water birds do (Randler, 2007), but walk between their nests and the beach many times in a breeding season. They step over or onto obstacles that they encounter along the way. Birds may use a dominant foot to step up or may use the dominant foot for balance and step up with the other foot (Harris, 1989; Tommasi & Vallortigara, 1999). We had no a priori reason to hypothesize a particular dominant foot in the population of Magellanic penguins.

Penguins spend much of their time on land prone, often lying outside the nest, and they extend one or occasionally both legs and feet for thermoregulation, particularly on hot days. Thermoregulatory postures involve trade-offs with predation risks (Anderson & Williams, 2010; Carr & Lima, 2012). Large mammals such as foxes (Lycalopex culpaeus), pumas (Puma concolor), smaller wild cats (Leopardus spp.), and elephant seals (Mirounga leonina) prey on adult penguins in continental colonies in South America (Clark & Boersma, 2006; Frere et al., 2010; Zanón Martínez et al., 2012), including Punta Tombo (D. Boersma, 1982–2018, personal observation). Penguins may have to get up quickly from a prone position when threatened by a predator or another penguin. Thermoregulation may be a more likely behavior to show a dominant foot than stepping up, as rising from a prone position likely requires more postural control (Tommasi & Vallortigara, 1999) than stepping up.

In addition, penguins may have more blood flow to one leg than to the other as in pigeons (Bernstein, 1974). It would be more efficient to extend the leg and foot with more blood flow in thermoregulation. The reproductive organs are on or more developed on the left side than the right side in birds, which may result in more blood flow to the right side (Sturkie, 1986). Gonads require a large amount of blood (Sturkie, 1986) and asymmetry in gonads may affect blood flow to the feet. In most species of birds, including Magellanic penguins, females have a single ovary and oviduct, on the left (Gill, 1995). In males of many species, the left testis is larger than the right (Birkhead, Fletcher & Pellatt, 1998). We hypothesized that penguins would extend the right leg and foot preferentially for thermoregulation on hot days.

Dominant flipper

Penguins use their flippers for swimming and may have a dominant flipper because of the lack of support water provides for sudden turns (reaction forces) (MacNeilage, 2014). Schooling fish, the main prey of penguins (Frere, Gandini & Lichtschein, 1996; Wilson et al., 2005), tend to turn one direction more than the other (Bisazza et al., 2000; Domenici, 2010). Penguins are opportunistic foragers, eating many species (Boersma et al., 2013; Frere, Gandini & Lichtschein, 1996), and the optimal dominant flipper for penguins may depend on the prey species if one species tends to escape in one direction and another species escapes in the other direction. Penguins often forage in groups, although they also forage alone (Berlincourt & Arnould, 2014; Gómez-Laich, Yoda & Quintana, 2018; Jehl, 1974; McInnes et al., 2017; Sutton, Hoskins & Arnould, 2015; Wilson & Wilson, 1990). We expected lateralization of individuals but not in the population.

A dominant flipper may result in structural asymmetries in the skeleton (MacNeilage, 2014; Shaw & Stock, 2009) and in asymmetrical stresses on the feathers leading to excessive feather wear on one flipper (Fallow et al., 2009). We assumed that the curved keels (carina) on the sterna that we observed on penguin carcasses indicated a dominant flipper, as bones respond over time to mechanical forces (Currey, 2003). Known juvenile carcasses were rare. Juveniles spend less time in the colony than adults do, as Magellanic penguins first breed at 4 years of age or older (Boersma et al., 2013; Boersma, Stokes & Yorio, 1990).

Preferred fight orientation

Penguins fight during the breeding season (Renison, Boersma & Martella, 2002, 2003; Renison et al., 2006) and may show evidence of lateralized brains in their preferred fighting orientation. The right hemisphere dominates in aggression (Bisazza, Rogers & Vallortigara, 1998) and many animals use their left eyes preferentially in aggressive interactions (Hews & Worthington, 2001; Rogers, 2008). Because fighting is a social behavior it may be more likely than individual behaviors to show lateralization in the population (Bisazza et al., 2000; Ghirlanda & Vallortigara, 2004; Rogers, Frasnelli & Versace, 2016). We expected penguins to use their left eyes more often than their right eyes in fights.

We looked for evidence of brain lateralization in aggressive behavior by observing the distribution of blood on penguins’ faces following fights, because it is difficult to directly observe a side preference in fights. Fights are uncommon, last only 2 or 3 mins on average, often occur inside nests (Renison et al., 2006), and are difficult to observe or film from the beginning. Magellanic penguins, both males and females, fight over nests and frequently suffer cuts to the bare skin of the face (Renison, Boersma & Martella, 2002, 2003; Renison et al., 2006), which might show lateralization.

We tested whether penguins display behavioral asymmetries in limb and eye use, mostly in the population, but in one case within individuals. We predicted that (1) equal numbers of penguins would step up onto an obstacle with the right and left legs (no lateralization of the population), but (2) more penguins would extend their right legs than their left legs during thermoregulation because of circulatory asymmetries (lateralization of the population). We further predicted that (3) individual penguins would have a dominant flipper, resulting in morphological asymmetries, but left- and right-flippered penguins would be equally likely in the population (individual, but not population lateralization). Finally, we predicted that (4) penguins would use their right hemispheres and left eyes in aggressive interactions, and would be bloodier on the left side than on the right side following fights (population lateralization).

Materials and Methods

Study site and species

Our study was part of a long-term (beginning in 1982) study of Magellanic penguins at Punta Tombo, Argentina (44°03′S, 65°13′W) (Boersma, Stokes & Yorio, 1990). Punta Tombo is a large but declining breeding colony, with 201,000 active nests in 2014 (Rebstock, Boersma & García-Borboroglu, 2016). It lies in arid coastal Patagonia, where breeding-season air temperatures exceed 35 °C in most years (Boersma & Rebstock, 2014; Stokes & Boersma, 1998).

Magellanic penguins breeding in Argentina are migratory, spending winters at sea off northern Argentina, Uruguay, and southern Brazil (Pütz, Ingham & Smith, 2000; Stokes et al., 2014), at least 1,000 km north of Punta Tombo. Foraging trip distance during breeding depends on the stage of the breeding season, and ranges from a few kilometers to >600 km (Boersma & Rebstock, 2009). Magellanic penguins feed in the ocean, usually on small schooling fish and squid, and sometimes crustaceans and jellyfish (Frere, Gandini & Lichtschein, 1996; Thiebot et al., 2017; Wilson et al., 2005). They often feed in small groups and in association with other seabirds and marine mammals (Gómez-Laich, Yoda & Quintana, 2018; Jehl, 1974; Wilson & Wilson, 1990), and use their bills for prey capture and handling.

Penguins are adapted to swimming in cold water, and warm temperatures on land cause heat stress (Boersma, 1975; Boersma & Rebstock, 2014; Frost, Siegfried & Burger, 1976). Magellanic penguins nest primarily in burrows or under bushes, where they are protected from the sun (Rebstock, Boersma & García-Borboroglu, 2016; Stokes & Boersma, 1991). While breeding, they spend time on land, resting, incubating eggs, and brooding or guarding chicks. Birds lose heat mostly through their bare skin, as feathered surfaces are too well insulated for heat loss. Legs and feet dissipate heat well (Baudinette et al., 1976; Drent & Stonehouse, 1971). On warm days, Magellanic penguins often lie with one or both feet out with the soles up and exposed to air, rather than tucked under the body. Such a posture may increase the risk of predation if rising takes longer than when both legs are under the body (Anderson & Williams, 2010; Carr & Lima, 2012).

We divided the colony into 16 areas where we walked slowly to make observations. The areas were sampled every 5–10 days. Field work was approved by the Institutional Animal Care and Use Committee of the University of Washington (protocol #2213-02), and the Province of Chubut, Argentina (Ministerio de Turismo and Ministerio de Desarrollo Territorial y Sectores Productivos, Subsecretaria de Agricultura y Ganadería, Dirección de Fauna y Flora Silvestre, 1194/16-MT).

Analyses

We used binomial (sign) tests with p = q = 0.5 to test whether our findings differed significantly from a 0.5:0.5 ratio (Zar, 1984). We also used χ2 tests in a few cases, noted below. We used two-tailed tests and considered p < 0.05 significant.

Dominant foot

One of us (TS) watched 300 penguins walking over obstacles (scales) and recorded the number of penguins that stepped up with each leg, in December 2014 and January 2015. The scales were set on the ground in a dry stream bed that penguins use as a path to travel between their nests and the sea. The scales were in custom-built boxes, 64.8 × 32.4 cm, 5.7 cm high, and covered with thick brown vinyl (Fig. 1). We counted only penguins that stopped and then stepped up onto a scale and excluded penguins that continued to walk. Penguins, like people, alternate feet when walking, so if they did not stop, the foot they used might depend on which foot was closest to the scale when they stepped up to cross, and would not reflect any preference. Although penguins with strong lateralization may adjust step length to step with the dominant foot, using only penguins that stopped should be a more sensitive test as penguins with weak lateralization would be less likely to use the dominant foot if they kept walking. Because of the large number of penguins crossing the scales and the short time we watched, we assumed we counted each penguin only once. During the days we watched penguins crossing, the scales recorded 1,272–4,321 weights each day.

Figure 1 Magellanic penguins crossing obstacles (scales) in their path.

We recorded whether a penguin that stopped then stepped onto the scale with its right or left foot. Photo credit: Clayton Gravelle.

We observed 121 penguins using their feet in thermoregulation as we walked around the colony December 1–17, 2014 and November 30 to December 20, 2015 (TS and PDB) and another 232 penguins from December 15, 2018 to January 4, 2019 (PDB) (Fig. 2). In 2014 and 2015, we searched for penguins using a foot in thermoregulation mostly on hot days, mostly in the middle of the day, when we were most likely to see this behavior. In 2018, to increase the sample size and see if the pattern held throughout the day, we recorded data in the morning or late afternoon, but not in the middle of the day. When penguins extend a foot, they usually keep the same foot out, sometimes for an hour or more. We recorded if a prone penguin had one or both feet out, and if only one foot was out, which one. Most penguins were unmarked, and not in study nests. It is unlikely that we counted the same penguin more than once because there are a large number of penguins in the colony and we counted penguins in any one area only once per day.

Figure 2 A Magellanic penguin using its extended right foot for thermoregulation on a hot day.

Photo credit: Dee Boersma.

We looked for asymmetry in male testes sizes because asymmetric testes may cause circulatory asymmetries, which may, in turn, make one foot more effective for thermoregulation than the other foot. We necropsied 55 adult males found freshly dead in the colony from 2002 to 2016 and measured the length of their left and right testes. We compared the left and right testes lengths using a paired t-test.

Dominant flipper

We looked for morphological evidence of a dominant flipper by examining the keel curvature and feather wear on the trailing edges of flippers because we could not directly observe penguins foraging (swimming) in the ocean. We collected intact sterna from the remains of dead penguins anywhere in the colony or adjacent beaches in 2014 and 2015. When the sternum was disarticulated from the rest of the skeleton without any plumage, we assumed the penguin was an adult when it died, although we did not know the sex or age of the individual. We found 76 sterna large enough to be from adults. In addition, we found nine carcasses with enough plumage remaining to determine that the penguins were juveniles when they died (<1.5 years of age, lacking the black and white bands and black faces of adults). We necropsied nine chicks 30–60 days of age that died of natural causes to obtain their sterna. We recorded the direction of curvatures in the keel: towards the penguin’s right, left, or both (double curve), or none (Fig. 3). Two people (TS and PDB) examined the keels together. We collected the sterna to prevent counting the same one more than one time.

Figure 3 Curvature of keel (carina) on the sterna of Magellanic penguins.

Anterior is toward the top of the figure. (A) Two sterna of presumed adult penguins. The keel on the left sternum has a curve to the penguin’s right; the keel on the right sternum curves to the penguin’s left. (B) A juvenile sternum as evidenced by the plumage remaining on the carcass. The keel lacks a curve. Photo credit: Dee Boersma.

A dominant flipper might also result in excessive wear of feathers on one side, if stresses on the flippers during swimming are not equal. We examined the feathers on the trailing edges of both flippers on 1,217 live unbanded adults from October 2008 through February 2018. We used only unbanded penguins because bands caused or worsened feather wear in little penguins Eudyptula minor (Fallow et al., 2009). Unworn flippers in Magellanic penguins have a narrow band of white feathers on the trailing edges (Fig. 4A). In worn flippers, the white feathers are irregular or missing (Fig. 4B). We did not include penguins if missing feathers appeared to result from an injury rather than wear. We banded 482 of these penguins after measuring them. The rest had 2 × 10-mm numbered tags in the webbing of the foot (Boersma & Rebstock, 2010), that are unlikely to affect feather wear on flippers. We measured each penguin one time.

Figure 4 Feather wear on the trailing edges of adult Magellanic penguins’ flippers.

(A) Normal white feathers on trailing edge of flipper. (B) Worn feathers on trailing edge of flipper. Photo credit: Dee Boersma.

We coded feather wear for each penguin for each flipper as: 1 = little or no wear, 2 = moderate wear, 3 = severe wear. Several people scored feather wear between 2008 and 2018, but for each penguin, the same person scored the left and right flippers, and new field workers were trained by PDB or GAR each year. We counted the number of penguins with the same wear on each flipper, the number with more wear on the left flipper, and the number with more wear on the right flipper. We used a binomial test to determine whether left flippers or right flippers had more wear. We also did a likelihood-ratio χ2 (G-test) to determine if left flipper wear and right flipper wear were independent of each other, and explored the patterns if they were not. Finally, we used a likelihood-ratio χ2 to determine if symmetry in flipper wear was independent of sex. We recorded the sex of 1,071 of the penguins with feather-wear scores, sexing penguins by their bills, which are deeper in males than in females (Boersma et al., 2013). We used the likelihood-ratio χ2 because 33.3% and 40% of cells in the two tests had expected values <5.

Preferred fight orientation

We predicted that penguins fighting beak to beak would use their left eyes preferentially and hence peck and bite their opponents’ left sides more than the right sides, resulting in more blood on the left side of the face. One person (TS) observed fighting penguins, and penguins that had recently fought, opportunistically while walking through the colony, visiting each study area every 5–10 days. Dried blood disappeared after about 5 days on penguins that remained on land and even sooner if a penguin went to the ocean to bathe. It is unlikely that we sampled the same penguin twice. Renison (2000) and Renison, Boersma & Martella (2003) found that between 15% and 30% of penguins fought more than once. We occasionally made video recordings of penguins fighting, and referred to those videos to test our assumption that they fight mostly beak to beak. We saw 215 penguins with blood on their faces from fights. We divided the penguin’s face into thirds and noted whether there was blood in one section (little blood), two sections (moderate blood), or all three sections (a lot of blood), and whether there was blood on only the right side, only the left side, or both sides (Fig. 5). In addition to a binomial test to compare right and left sides of the face, we used a χ2 test to determine if the side of face bloodied was independent of the amount of blood.

Figure 5 Scoring of blood on the face of a Magellanic penguin that recently fought another penguin.

We scored blood on up to 1/3 of the face as “little,” from 1/3 up to 2/3 of the face as “moderate,” and more than 2/3 of the face as “a lot” of blood. We also noted whether there was blood only on the right side, only on the left side, or on both sides. Photo credit: Dee Boersma.

Results

Dominant foot

We did not find a side preference for stepping up onto an obstacle (n = 300 penguins). Penguins were equally likely to step up with their left (n = 140 penguins, 47%) or right (n = 160 penguins, 53%) foot (p = 0.27).

Fewer penguins extended the left foot for thermoregulation than the right foot in 2014 and 2015. We found no preference in which foot was extended in 2018–2019. The mixed results may be because the time of day we searched differed between the last season and the first two seasons (see Discussion). In 2014–2015, 39 penguins extended only the left foot, 78 extended only the right foot (p < 0.001), and four penguins extended both feet (Fig. 6A). In 2018–2019, we found no preference in the foot extended: 115 penguins extended only the left foot, 106 extended only the right foot (p = 0.59), and 11 penguins extended both feet (Fig. 6B).

Figure 6 Magellanic penguins used the right foot preferentially in thermoregulation in two seasons, but not in a third season.

(A) Twice as many penguins extended the right foot for thermoregulation as the left foot on hot days in the 2014 and 2015 breeding seasons. Four penguins extended both feet. (B) Approximately half of the penguins in the 2018 breeding season extended the right foot and half the left foot. A total of 11 penguins extended both feet. We recorded data on all days that we saw thermoregulation behavior, and only in the mornings and late afternoons.

Left testes averaged 3.19 ± 1.19 cm (mean ± standard deviation) and right testes averaged 2.54 ± 0.99 cm (n = 55). Left testes were 26% longer than right testes (t54 = 8.4, p < 0.0001).

Dominant flipper

The sterna of chicks were unossified and the keels were straight (n = 9). The sterna of juveniles were ossified and the keels were straight (n = 9). Of 76 presumed adult sterna, 46 (60.5%) keels curved one direction or the other, 5 (6.5%) keels curved in both directions (double curves), and 25 (33%) keels were straight (Fig. 7). Of the keels that curved one direction, left and right curves were equally likely (22 left, 24 right, p = 0.88).

Figure 7 More than half of individual Magellanic penguins showed morphological evidence of a dominant flipper, but left and right dominants were equally likely in the sample.

We collected sterna from penguin carcasses at Punta Tombo and examined the keels for curvature. A total of 29% curved to the penguin’s left, 31.5% curved to the penguin’s right, 6.5% had curves in both directions, and one-third had no curve (n = 76).

Feather wear on the trailing edges of flippers was uncommon, especially in males. We found feather wear in 179 of 1,217 unbanded penguins (15%). Of the penguins with feather wear, 45 had equal wear on both flippers. The remaining 134 penguins (11%) were equally likely to have more wear on the left (n = 74) or the right (n = 60) flipper (p = 0.26).

Penguins were more likely than expected to have no feather wear or to have equal feather wear on both flippers (diagonal elements in Table 1; G4 = 249.7, p < 0.001). Wear on one flipper predicted to some extent wear on the other flipper. Penguins with little or no wear on one flipper were less likely than expected to have moderate or severe wear on the other flipper, and penguins with severe wear on one flipper were more likely than expected to have moderate or severe wear on the other flipper (Table 1).

Table 1 Feather wear on the trailing edges of Magellanic penguins’ flippers was uncommon and was equal on the right and left flippers more often than expected (diagonal elements).

		Right flipper	
Little or no wear	Moderate wear	Severe wear	
Left flipper	Little or no wear	1038 (980.7)	47 (77.0)	5 (32.2)	
Moderate wear	48 (70.2)	22 (5.5)	8 (2.3)	
Severe wear	9 (44.1)	17 (3.5)	23 (1.4)	
Note:

Observed frequencies are listed first in each cell, followed by expected frequencies in parentheses. Observed frequencies that are greater than expected are in bold type; those that are less frequent than expected are italicized.

Females were twice as likely as males to have feather wear on either or both flippers (χ2(1) = 17.4, p < 0.001). A total of 19% of females had some feather wear compared to 10% of males (Fig. 8A). Females were more likely than males to have more wear on one flipper than on the other flipper (Fig. 8B; G4 = 15.1, p = 0.004). Proportionately more females (13.7% of 541) than males (8.1% of 530) had asymmetrical wear, but among penguins with some wear, proportionately more males (80% of 54) than females (71% of 104) had asymmetrical wear (Fig. 8B). Of the 74 females with asymmetrical wear, 41 had more wear on the left and 33 had more wear on the right (Fig. 8B; p = 0.42). Of the 43 males with asymmetrical wear, 22 had more wear on the left and 21 had more wear on the right (Fig. 8B; p ∼ 1.00).

Figure 8 More female Magellanic penguins than males had feather wear on the trailing edge of the flippers, and more females (n = 74) than males (43) had asymmetrical wear (more wear on one flipper than on the other).

(A) More females than males showed feather wear on their flippers. Penguins replace their feathers annually, so noticeable feather wear is uncommon, especially in males. (B) Of the penguins with some feather wear, right and left flippers were equally likely to have more wear (males: p ∼ 1.00, females: p = 0.42).

Preferred fight orientation

Contrary to expectations, penguins were more likely to be bloody on the right side of the face than on the left side (Fig. 9). A total of 40 penguins had blood on both sides. Less than half as many penguins had blood only on the left side (n = 55) as only on the right side (n = 120, p < 0.001). The distribution of blood on the face depended on the amount of blood (Fig. 9; χ2(4) = 10.7, p = 0.03). Of the penguins with blood on only one side, the tendency to have blood only on the right increased with the amount of blood. Penguins with only a little blood were equally likely to have it on either side (p = 0.20). Penguins with moderate amounts of blood were over three times more likely to have it only on the right than only on the left (p < 0.001). Penguins with a lot of blood were almost five times more likely to have it only on the right (p = 0.002).

Figure 9 Magellanic penguins that had recently fought other penguins were more likely to have blood only on the right side of the face than only on the left side.

The tendency to have blood only on the right side or on both sides increased as the amount of blood increased. Penguins with blood covering less than 1/3 of the face were scored as “little blood”; those with blood covering more than 2/3 of the face were scored as “a lot of blood.”

Penguins pecked and bit while facing each other, with their heads side by side facing the same direction, and at every angle in between (Fig. 10). In any orientation except directly face to face, if the more aggressive penguin looks with its left eye, it sees the right side of the other penguin’s face. Our observations and videos of penguin fights showed fighting directly face to face was not always the dominant orientation. Assuming the more aggressive penguin gets less blood on its face than the penguin it attacks, the observed pattern should be more blood on the right side of the face (Fig. 10), as we found.

Figure 10 Fight orientations in Magellanic penguins.

There is a continuum from 1 to 4. The bird labelled A in each pair is the aggressor. The hatched area represents the blood resulting from cuts by the opponent’s bill. In 2–4, the aggressor views his opponent primarily with his left eye; in 1, the birds view each other with their right eyes. In the course of a fight that lasts more than a few seconds, all these orientations could occur. Magellanic penguins had blood only on the right side twice as often as only on the left side, suggesting orientations 2–4 predominate.

Discussion

Lateralization in Magellanic penguins depended on the behavior we observed. We found evidence for dominant flippers in individual penguins, but not in the population; approximately half the penguins were right sided and half left sided. We could not test for individual preferences in the other behaviors (stepping up, thermoregulation, and fights) as we only had one observation per penguin. We did not find a dominant foot in stepping up for the population. We found a significant bias in foot use for thermoregulation in 2014 and 2015, likely because we concentrated our effort during the heat of the day. When we looked for extended feet in 2018–2019, we did not find a preference, likely because we looked early and late in the day. We found lateralization in fight behavior, the only social behavior we tested; the left eye was used for aggression, consistent with other vertebrate species.

Dominant foot

Penguins at Punta Tombo walk up to a kilometer to transit between their nests and the beach (Rebstock, Boersma & García-Borboroglu, 2016; Walker, Boersma & Wingfield, 2004), and encounter many obstacles on land, including rocks, branches, and burrow entrances or depressions in the ground. We found no dominant foot for stepping up onto an obstacle in the population. It is possible that individual penguins have a dominant foot, but approximately half are right footed and half left footed.

Stepping is not always a reliable indicator of a side preference. Stepping up is an unskilled task; skilled tasks (e.g., kicking a ball) are more likely to show a dominant foot in humans (Schneiders et al., 2010). Fewer than half of 32 dogs showed a paw preference in stepping down a step in one study (Wells et al., 2018), but the front paw dogs first used in stepping after standing still was a reliable indicator of a dominant paw in another study (Tomkins, Thomson & McGreevy, 2010).

We predicted that we would find a preferred foot used in thermoregulation in the population because of circulatory asymmetries. If one foot receives more blood than the other foot, that foot would be more efficient to use for thermoregulation. Well-developed ova and the shell gland, on the left side of female birds, use a lot of blood (Sturkie, 1986). As in many bird species, we found that the testes of breeding male Magellanic penguins are more developed on the left side than on the right side. The right foot may receive more blood than the left foot because the reproductive organs are on the left side. Asymmetries occur in the circulatory systems of birds (Odlind, 1978; Porter & Witmer, 2016), including blood flow to the feet (Bernstein, 1974).

We found a preference to extend the right foot in 2014 and 2015 but no preference in 2018–2019. We suggest that temperature explains the mixed results because we looked for penguins thermoregulating specifically during hot periods in 2014 and 2015, but during mornings and evenings in 2018–2019. How hot the penguin is may influence which foot is extended. If circulatory asymmetries drive which foot is extended, penguins should use the most efficient foot, or both feet, on the hottest days. Body temperature in African penguins Spheniscus demersus was higher on hot sunny days than on cool overcast days, and higher in the middle of the day than early morning or evening (Frost, Siegfried & Burger, 1976). It may not matter which foot a penguin uses for thermoregulation on cooler days or at cooler times of day, when its body temperature is lower and it does not need to dump heat rapidly.

Although extending both legs and feet would be even more efficient for losing heat, this was rare. Having both feet extended would likely increase the time it takes a penguin to stand up much more than extending only one leg. It also may be more effective to stand and pant when very hot (Boersma, 1975).

Dominant flipper

We found evidence that some individual penguins have a dominant flipper. Asymmetry in wings is maladaptive in flying birds and may result from fluctuating asymmetries or imperfect bilateral development (Balmford, Jones & Thomas, 1993; Brown & Bomberger Brown, 1998). Penguins “fly” through the water using their flippers to counteract their positive buoyancy when descending during a dive (Mattern et al., 2018). The advantages of flipper asymmetry in water, a dense medium, may outweigh the disadvantages. MacNeilage (2014) hypothesized that side biases are common in marine mammals because water does not provide support for the reactive component in movement. A sudden change in movement, such as a turn, requires support for the reaction forces (“equal and opposite reaction”). For example, a running person pushes against the ground with the left foot to turn right. When a swimming animal makes a sudden turn, the water provides little resistance. Air also does not support the reaction forces, but water is much denser than air and resists sudden movements more than air (MacNeilage, 2014). Hence, morphological asymmetries in the keels and flippers may help penguins manoeuver more efficiently in the water.

We assumed that the behavior (swimming with a dominant flipper) was the cause rather than the effect of the morphological asymmetries because we found the keel asymmetries in adults, but not in chicks or juveniles. The curve likely results from uneven forces on the bone (Currey, 2003; Shaw & Stock, 2009). Assuming the pectoralis and supracoracoideus muscles on the dominant side are thicker (more muscle fibers) than those on the opposite side, the pull on the keel is asymmetric, resulting in uneven forces and bone remodeling. A curved surface is stronger than a flat surface, so the curve may help resist the unequal pull. The curve overlaps the section of the keel where the bone is thinnest (Fig. 11). The curvature also increases the surface area for attachment of muscles, without increasing the length or depth of the keel.

Figure 11 The curve in keels of Magellanic penguins’ sterna overlaps the area of thinnest bone.

The curve strengthens the keel against the uneven pull of the muscles, and increases the surface area for muscle attachment. Photo credit: Ginger Rebstock.

Keels on sterna that curved one direction (60.5%) were much more common than asymmetrical feather wear on the flippers (11%). Keel curvature likely accumulates over the life of the penguin. The sternum is among the last bones to ossify in birds, generally after hatching (Maxwell & Harrison, 2008; Mitgutsch et al., 2011), and the pectoralis muscles are relatively small until the chick begins exercising its wings prior to fledging, or begins flying (Bennett, 2008; Carrier & Leon, 1990). Before the pectoralis muscles develop fully, asymmetrical forces on the keel would not be great enough to cause remodeling or deformation of the bone. Magellanic penguin chicks exercise their flippers by rapidly flapping them only a few weeks before fledging. Curved keels were not present in juveniles suggesting more than 1 year at sea is needed for a keel to accumulate a noticeable curve. Feather wear on flippers, in contrast, accumulates over 1 year and then is eliminated following the molt as the vast majority of penguins molt annually, replacing worn feathers (Boersma et al., 2013). The time for the side bias to affect the feathers is only 1 year, relatively short compared to the lifetime of the bird for the keel curvature.

Females were more likely than males to have feather wear and asymmetrical feather wear on their flippers. Females are smaller than males on average (Boersma et al., 2013) and have a narrower isotopic niche width (eat fewer species of prey) (Silva et al., 2014). Smaller penguins cannot dive as deeply or as long as larger penguins (Walker & Boersma, 2003). Overwinter mortality is higher in females than in males (Vanstreels et al., 2013), especially in years when overall mortality is high and prey availability is presumably low (Gownaris & Boersma, 2019). Hence, females likely work harder than males when foraging, leading to greater feather wear on females’ flippers.

Preferred fight orientation

We found evidence for a preferred fighting orientation in Magellanic penguins, consistent with the more aggressive penguin in a fight using its left eye and attacking the right side of the opponent. The result was opposite our prediction because we overestimated the amount of time penguins fight facing each other (bill-to-bill) and did not take into account the greater amount of blood likely on the face of the less aggressive penguin compared to the more aggressive penguin. Animals in a wide range of taxa use the left eye (or left antenna) preferentially in aggressive interactions with conspecifics (Bisazza, Rogers & Vallortigara, 1998; Hews & Worthington, 2001; Rogers, Frasnelli & Versace, 2016).

Penguins’ fight behavior is also consistent with evidence that lateralization is more likely in a population in social contexts than in individual contexts (Ghirlanda, Frasnelli & Vallortigara, 2009; Ghirlanda & Vallortigara, 2004; Rogers, Frasnelli & Versace, 2016). Aggressive interactions are more strongly lateralized than nonaggressive interactions in many species (Deckel, 1995; Rogers, Frasnelli & Versace, 2016).

Asymmetries in circulation could contribute to the pattern we found. In double-crested cormorants (Phalacrocorax auritus), the left palatine artery is larger than the right palatine artery (Porter & Witmer, 2016). If an artery supplying the right side of the face is larger than the corresponding artery on the left in Magellanic penguins, they would bleed more on the right when cut.

The percentages of penguins with blood on both sides of the face and only on the right side of the face increased as the amount of blood increased. We would expect to see more blood on both sides, as the more blood present, the less likely it is to be concentrated in a small area of the face. The proportion of penguins with blood only on the right side increased with the amount of blood. This pattern is consistent with the tendency to use the left eye more as aggression increases (Deckel, 1995), and more blood likely indicates more aggression and more intense fights. Magellanic penguin fights that lasted longer and were in higher quality nests resulted in more cuts to the face (Renison et al., 2006).

Conclusions

Magellanic penguins join the growing list of species that show evidence of lateralization in the wild. The lateralization depended on the behavior or context, and was sometimes evident for individuals, but not for the population. Penguins lack the behaviors typically tested in birds to look for a dominate foot, such as standing on one foot or manipulating food with a foot (Randler, 2007; Rogers, 2009). We found no lateralization for the population when stepping up. Population lateralization in the use of a foot for thermoregulation was sometimes evident. We suggest that penguins show lateralization at hotter temperatures but not when cooler. We found over half of adult Magellanic penguin keels showed morphological evidence of a dominant flipper, but the population did not show any flipper preference. Stepping up, thermoregulating, and swimming are individual behaviors. In fights, the only social behavior we tested, attacking Magellanic penguins are likely to use the left eye, resulting in more blood on the right side of the face of the opponent (the less aggressive penguin). The preference for using the left eye in aggressive interactions is found in many other species (Bisazza, Rogers & Vallortigara, 1998; Rogers, 2008) and shows specialization of the brain hemispheres (Rogers, 2017). Our results are also consistent with other studies showing lateralization of a population is more likely in social contexts (Vallortigara & Rogers, 2005). Like many other species, Magellanic penguins show context-specific lateralization.

Supplemental Information

Supplemental Information 1 Magellanic penguins found freshly dead at Punta Tombo, Argentina, were necropsied and their gonads measured.

Raw data from 2002 to 2016.

Click here for additional data file.

Supplemental Information 2 Counts of Magellanic penguins that had recently fought other penguins, by the amount and location of blood on their faces.

Counts from Punta Tombo, Argentina, 2014-2015. Columns: Right = blood only on the right side of the face; Left = blood only on the left side of the face; Both = blood on both sides of the face. Rows: Little = blood covering less than 1/3 of the face; Moderate = blood on more than 1/3 and less than 2/3 of the face; A lot = blood covering more than 2/3 of the face.

Click here for additional data file.

Research at Punta Tombo was carried out under a joint agreement between the Wildlife Conservation Society and the Office of Tourism, Province of Chubut, Argentina. We thank the Province of Chubut and particularly the Offices of Tourism and of Flora and Fauna, and the National Scientific and Technical Research Council of Argentina for permits and/or logistical support. The La Regina family allowed access to the penguins breeding on their land. We thank the La Regina family, Carlos García, William Conway, Graham Harris, Patricia Harris, Carol Passera, Carlos Passera, and Centro Nacional Patagónico, Argentina (CENPAT), and key wardens for logistical support and help over the 35 years of study. We are grateful to the many students and field volunteers who collected data. Discussions with Adam Summers helped clarify the predictions and their mechanisms. Gordon Orians, Eric Wagner, and two anonymous reviewers improved the manuscript.

Additional Information and Declarations

Competing Interests

Author Contributions

Animal Ethics

Field Study Permissions

Data Availability

Ginger Rebstock and Dee Boersma are Academic Editors for PeerJ.

Thaís Stor conceived and designed the experiments, performed the experiments, analyzed the data, prepared figures and/or tables, authored or reviewed drafts of the paper, approved the final draft.

Ginger A. Rebstock analyzed the data, prepared figures and/or tables, authored or reviewed drafts of the paper, approved the final draft.

Pablo García Borboroglu authored or reviewed drafts of the paper, approved the final draft, obtained permits for field work.

P. Dee Boersma conceived and designed the experiments, performed the experiments, authored or reviewed drafts of the paper, approved the final draft.

The following information was supplied relating to ethical approvals (i.e., approving body and any reference numbers):

The Institutional Animal Care and Use Committee of the University of Washington approved the research (IACUC Protocol Number 2213-02).

The following information was supplied relating to field study approvals (i.e., approving body and any reference numbers):

Field research was approved by the offices of Tourism and of Flora and Fauna, Chubut, Argentina (001194/16-MT).

The following information was supplied regarding data availability:

The raw data are presented in the text or available in the Supplemental Files.

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
