# Peer review of "Lateralization (handedness) in Magellanic penguins"

_PeerJ, doi:10.7717/peerj.6936_

## Round 0.1 · original submission · Major Revisions

Please respond to all of the points raised by the reviewers.

Reviewer 1 ·

Basic reporting

See below

Experimental design

See below

Validity of the findings

See below

Comments for the author

I believe this is a good paper, that extends in a significant way our knowledge on lateralization in vertebrates in a species that so far has been little investigated. The observations appeared to be well-conducted and the data properly analyzed.
I noticed, however, that the authors do not properly managed all the literature (particularly the most updated one), and I make several suggestions on this below.
As to the lack of footedness for stepping and its presence in thermoregulation, I am wondering what type of challenge the thermoregulatory posture would pose to the animals. Apparently little because they were prone. Nonetheless I am wondering whether the extension of one leg makes the other leg forced to play an important role in the control of posture. I am asking this because in other birds species it has been argued that lateralization in foot use maybe a consequence of the foot that is actually used to control posture (see e.g. Tommasi, L., Vallortigara, G. (1999). Footedness in binocular and monocular chicks. Laterality, 4: 89-95; Versace, E., Vallortigara, G. (2015). Forelimb preferences in human beings and other species: multiple models for testing hypotheses on lateralization. Frontiers in Psychology, 6: 233. doi: 10.3389/fpsyg.2015.00233).
I found the interpretation for fight behaviour quite convincing.


Line 45: Please add here also more updated reviews:

https://www.cambridge.org/core/books/divided-brains/442F89DBA8F4BA76CD2F896E960A1E36

https://www.scientificamerican.com/article/evolutionary-origins-of-your-right-and-left-brain/

https://onlinelibrary.wiley.com/doi/abs/10.1002/wcs.100

http://psycnet.apa.org/record/2017-00569-027

https://www.sciencedirect.com/science/article/pii/S0093934X00923034


For evidence in invertebrates see e.g.

https://r.unitn.it/filesresearch/images/cimec-abc/Publications/2012/frasnelli_neuro-biobe.pdf

https://www.nature.com/articles/srep29411

https://journals.plos.org/plosone/article?id=10.1371/journal.pone.0002340

http://rspb.royalsocietypublishing.org/content/282/1803/20142571

https://www.sciencedirect.com/science/article/pii/S0960982218309187?dgcid=rss_sd_all



Line 47:
https://www.cambridge.org/core/journals/behavioral-and-brain-sciences/article/survival-with-an-asymmetrical-brain-advantages-and-disadvantages-of-cerebral-lateralization/2E4A2B275EED0B6C037FCC84C62F6F7D

https://www.cambridge.org/core/journals/behavioral-and-brain-sciences/article/forming-an-asymmetrical-brain-genes-environment-and-evolutionarily-stable-strategies/0617C9F89F054C0B4CE2537ABCF7303E

https://www.ncbi.nlm.nih.gov/pmc/articles/PMC1691668/

http://rstb.royalsocietypublishing.org/content/364/1519/861.article-info

https://onlinelibrary.wiley.com/doi/abs/10.1002/dev.20166


Line 56: here a more general view of the functions of the left and right hemispheres should be provided (including e.g. space, emotions, motivation etc). Please refer to the general reviews I suggested above.

Line 66: see however evidence above (suggestions for line 47) concerning the role of social interaction in the alignment of lateralization (see also


Line 79: This statement is obscure to me. Perhaps the authors refer to frequency-dependent selection for asymmetries see above e.g. https://onlinelibrary.wiley.com/doi/abs/10.1002/dev.20166


Line 346 see also

https://www.researchgate.net/publication/8443269_Paw_preference_in_dogs_Relations_between_lateralised_behaviour_and_immunity

https://www.sciencedirect.com/science/article/pii/S0304394010004532

https://www.ncbi.nlm.nih.gov/pubmed/17884191

Reviewer 2 ·

Basic reporting

Literature survey: this part could be improved while increasing the focus of the paper. For instance: is this paper about functional or morphological asymmetries? Individual or population asymmetries? Social or individual behaviours?

In particular, there is no distinction between individual and population-level asymmetries, although this is a crucial distinction for this paper. Moreover, the authors often shift between morphological and functional asymmetries in different traits. Sometimes there are cases overgeneralisation/oversimplifications; some examples: "Brain lateralization is considered adaptive"...this is a bit vague. "Brain lateralization is thought to be adaptive, especially in social or group behaviors"... what about lateralisaton in individual behaviours such as gap crossing in locusts?; "Brain lateralization appears to enhance cognitive ability by allowing both brain hemispheres to specialize and function simultaneously (Rogers 2017) and is often linked to behavioral lateralization, or to hand, foot, or paw preferences (Rogers 2009)." ... there are different explanations for the advantages of different types of lateralisation. The cited references are valuable but it would be better to refer also to the primary literature; "The strength of lateralization may be more important than the direction within an individual" ...it is not clear in which sense "more important"?. ; "The left hemisphere (hence, the right side of the body) dominates in vocalization": I am not sure in which sense the right side of the body dominates in vocalisation; "The optimal dominant flipper may depend on the prey species." ... this is not clear. "A dominant side may be
advantageous when it is rare, leading to an equal ratio of dominant sides in the population)"... this is not clear.

Data should be presented in figures, e.g. bars that present frequency of different outcomes

Experimental design

Lateralisation is a huge topic, and focus and research questions are not defined.
There are some distinctions that should be stated more clearly about the focus of the paper: 1) individual vs population level lateralisation 2) functional vs morphological asymmetry 3) individual vs social behaviours 4) why this species is interesting to address the specific questions.
In this manuscript, the authors mainly look at population level lateralisation but for flipper asymmetries, they look at individual-level asymmetries; sometimes they look at behavioural asymmetries and sometimes at morphological asymmetries but the two measures are not related to the same trait.
Overall, the focus of the paper is less than clear: did Authors select specific traits for any theoretical reason? It should be stated whether the study is a preliminary descriptive observation or whether the traits investigated are theoretically connected. The impression from reading the paper is that this is a general survey of lateralisation in which the investigates traits are not theoretically connected to one another.
If besides the focus of the paper authors have also included other measures, these can be reported in a separate section.

Validity of the findings

Did Authors measure the inter-rater reliability between observations?

---

## Round 0.2 · Minor Revisions

One of the reviewers has suggested a number of changes to improve your paper. Please address all of the points raised by this reviewer.

Reviewer 1 ·

Basic reporting

I believe the authors have addressed satisfactorily my comments.

Experimental design

No comments

Validity of the findings

No further comments.

Comments for the author

I believe the authors have addressed satisfactorily my comments.

Reviewer 2 ·

Basic reporting

Mostly clear but some paragraphs should be moved to different sections (see below)

FIgures: to help comparisons between conditions with diffrent subjects it would be better ti plot proportions (adding counts in tables)

No strong conclusions on the role of temperature can be done since this variable was not directly analysed.

Experimental design

see below

Validity of the findings

No strong conclusions on the role of temperature can be done since this variable was not directly analysed. Speculation on the role of temperature should be identified as such.

Comments for the author

Abstract and introduction

24. “regardless of the species” : This sounds to me an oversimplification that implies that all the species exhibit the same pattern.
27 “side preferences”: side preferences might be different from lateralisation and be transitory.
29 and following: usually the abstract doed not contain p values and exact numbers

“Lateralization depended on the behaviour tested and, in thermoregulation, on the temperature”: actually the role of the temperature was not directly tested.

“solitary behaviour”: here and in other cases: since penguins were usually in groups, would it be more appropriate to use the term “individual behaviour/s”?

56-57: “is thought to be adaptive”: I would rephrase in a similar way as: "Several scholars hypothesise that… ( refs)".

163: it would be important to cite also the original papers about the role of social behaviour/aggression in individual vs social behaviour.

167-169: so that is a case of morphological, not of functional asymmetry.

Materials and methods
Throughout the materials and methods, there are parts that are more appropriate for the introduction: e.g. 215-218, 230-234, 247-250, 300-303 and in the dominant flipper part. A possibility to help the readers to find the information in the sections where it is usually located would be to dedicate paragraphs in the introduction to explain the relevance and details of each variable investigated.

208-211: the heading is about "Study site and species" but this part is about data analyses.

I understand that a binomial test with p=0.5 is a sign test, but for clarity I would use consistently the term sign test of binomial test throughout the manuscript instead of shifting between terms.

223-224: it is possible to automatically adjust the length of steps during walking, without stopping, so I don’t fully understand the rationale of considering only penguins that did stop.

** 237 and below: the Authors explain the inconsistency in feet preference between different year as an effect of temperature. Although this is a possible explanation, other variables not controlled by the experiments might have differed between different years of observations. The fact that the temperature was not analysed to observe whether it made a difference (especially within a year of observation), prevents to draw strong conclusions. So I would recommend modifying the abstract and discussion/conclusions accordingly.

247-252 I find a bit confusing that the paragraph about gonads (247-252) is included in the section "Dominant foot" (similarly in the Results section).

303-304: can the circulation asymmetry explain also the asymmetry in eye use?

In the Results (371), the Authors mention videos about the fights but there is no mention of these videos in the methods. It would improve clarity a description of when the videos where recorder and how (unless I have overlooked this detail)

Results

* For not significant results the authors use for instanc “p > 0.2” or “p> 0.5”. It would be more precise to specify the exact p level.




The Results section in research reports typically does not provide interpretation. I would suggest to revise the results section accordingly (see for instance 369-377).

Discussion

406-407: the Authors might want to clarify whether here they are referring to their own results

410 and following: it is important to clarify the limitations of this interpretation (or to provide direct evidence of the association between temperature and lateralisation)

464-469: It is not entirely clear to me the connection between the paragraph 464-469 and the rest of the discussion. Is this rather some information for the introduction? Or can the Authors make more clear the connection with their results?

480: it would be interesting to refer also to the original theoretical accounts.

483: “usually”: can the Authors specify in which contexts?

491-493: this would be more clear using a line plot to illustrate

503-505: it would be important not to make strong claims about the role of temperature is that has not been explicitly investigated (see my comments above)


Figures and captions:
- exact p levels are more informative also in the case of non-significant results

** Figure 8: it is extremely hard to compare counts when the overall number of subjects considered is different (especially in Figure 8B, with 54 mals and 104 females). The AUthors should consider plotting proportions or at least put side by side the bars that can be directly compared: (all data from males) and (all data from females)

** Figure 9: same considerations as above about the counts/frequencies. Plotting proportion and using a line plot would make much clearer how the proportion of use of one side changes with the increase of blood.

---

## Round 0.3 · accepted · Accept

I found your paper very interesting. It will make an important contribution to the study of lateralization in nonhuman species.

#